# Nuclear and mitochondrial DNA editing in human cells with zinc finger deaminases

Kayeong Lim [1,3], Sung-Ik Cho[1,2,3] & Jin-Soo Kim [1✉]

Base editing in nuclear DNA and mitochondrial DNA (mtDNA) is broadly useful for bio-medical research, medicine, and biotechnology. Here, we present a base editing platform, termed zinc finger deaminases (ZFDs), composed of custom-designed zinc-finger DNA-binding proteins, the split interbacterial toxin deaminase DddA$_{tox}$, and a uracil glycosylase inhibitor (UGI), which catalyze targeted C-to-T base conversions without inducing unwanted small insertions and deletions (indels) in human cells. We assemble plasmids encoding ZFDs using publicly available zinc finger resources to achieve base editing at frequencies of up to 60% in nuclear DNA and 30% in mtDNA. Because ZFDs, unlike CRISPR-derived base editors, do not cleave DNA to yield single- or double-strand breaks, no unwanted indels caused by error-prone non-homologous end joining are produced at target sites. Furthermore, recom-binant ZFD proteins, expressed in and purified from *E. coli*, penetrate cultured human cells spontaneously to induce targeted base conversions, demonstrating the proof-of-principle of gene-free gene therapy.

[1] Center for Genome Engineering, Institute for Basic Science, Daejeon 34126, Republic of Korea. [2] Department of Chemistry, Seoul National University, Seoul 08826, Republic of Korea. [3] These authors contributed equally: Kayeong Lim, Sung-Ik Cho. ✉email: jskim01@snu.ac.kr

A growing list of tools have been reported for genome editing in eukaryotic cells and organisms, a broadly useful method in research and medicine, which now include, but are not limited to, zinc finger nucleases (ZFNs)[1], transcription activator-like effector (TALE) nucleases (TALENs)[2], TALE-linked split interbacterial deaminase toxin DddA-derived cytosine base editors (a.k.a. DdCBEs)[3], CRISPR-Cas9[4–6], and catalytically-impaired Cas9-linked deaminases (a.k.a. base editors)[7–9]. These tools, in principle, are composed of two functional units: a DNA-binding moiety and a catalytic moiety. Thus, a zinc finger array or a TALE array functions as a DNA-binding moiety, whereas a nuclease (FokI in ZFNs and TALENS) or a deaminase enzyme (split-DddA$_{tox}$ in DdCBEs and APOBEC1 in CBEs) functions as a catalytic unit. CRISPR-Cas9 is both a nuclease enzyme and an RNA-guided DNA-binding protein. Custom-designed, programmable nucleases such as ZFNs, TALENS, and Cas9 cleave DNA, producing double-strand breaks (DSBs), the repair of which give rise to gene knock-out and knock-in in a targeted manner. Programmable nuclease-induced DSBs, however, can cause unwanted large deletions[10–13] at on-target sites, p53 activation[14–16], and chromosomal rearrangements[17,18] resulting from the repair of two concurrent DSBs at on- and off-target sites. In contrast, programmable deaminases, including cytosine and adenine base editors (CBEs[7,9] and ABEs[8]), do not produce DSBs, avoiding these unwanted events in cells, and catalyze single nucleotide conversions efficiently without a repair template or donor DNA. Note, however, that CBEs and ABEs, containing the Cas9 nickase variant, cleave the target DNA strand and produce nicks or single-strand breaks, which can still cause unwanted indels at target sites.

Recently, Mok et al.[3] demonstrated that the interbacterial deaminase toxin DddA$_{tox}$ derived from *Burkholderia cenocepacia* can be split and fused to TALE arrays and a uracil glycosylase inhibitor (UGI) to create DdCBEs, which catalyze C-to-T base conversions in nuclear DNA and mitochondrial DNA (mtDNA) in mammalian cells. We and others also showed mtDNA editing in mice[19] and chloroplast DNA editing in plants[20,21] using custom-designed DdCBEs. In this study, we sought to create zinc finger deaminases (ZFDs) for indel-free, precision base editing in human and other eukaryotic cells by linking split-DddA$_{tox}$ to custom-designed zinc finger proteins (ZFPs). Because zinc finger arrays (encoded in a $2 \times 0.3{\sim}0.6$ k base pair (kbp) DNA) are small in size, compared to TALE arrays ($2 \times 1.7{\sim}2$ kbp) or S. pyogenes Cas9 (4.1 kbp)[22], ZFD-encoding genes can readily be packaged in a viral vector with a limited cargo space such as AAV for in vivo studies and gene therapy applications. Unlike TALE arrays, zinc finger arrays lack bulky domains at both the C terminus and the N terminus, making them engineering friendly: split-DddA$_{tox}$ halves can be fused to either terminus of a ZFP. Furthermore, ZFPs with an intrinsic cell-penetrating activity[23–26] may allow nucleic acid-free gene editing in human cells. These properties may make ZFPs an ideal platform as a DNA-binding module for base editing in nuclear and organelle DNA.

## Results

**Optimization of ZFD architecture**. To develop ZFDs for base editing in human and other eukaryotic cells, we first sought to optimize both the length of the amino-acid (AA) linkers that connect ZFPs to split-DddA$_{tox}$ halves and the length of the spacers, where C-to-T conversions would be induced, between the left and right ZFP-binding sites. We chose a well-characterized ZFN pair, targeted to the human *CCR5* gene, to make ZFDs with variable linkers of 2, 5, 10, 16, 24, and 32 AA residues in length and constructed a series of target plasmids (pTargets) that contained left and right ZFP-binding sites separated by variable

spacers, composed of repetitive 'TC' motifs, of one to 24 base pairs (bps) in length (Fig. 1a, b and Supplementary Table 1, 2). Note that DddA$_{tox}$ can be split at two positions (G1333 and G1397) and that each half can be fused to either a left or right ZFP. We measured the base editing efficiencies of the resulting 24 (= 6 linkers × 2 split positions × 2 possible fusions (left or right)) ZFD constructs with each of the 24 pTarget plasmids in HEK 293 T cells at day four post-transfection via targeted deep sequencing. ZFD constructs with short linkers (2- and 5-AA in length) were poorly efficient or inactive, whereas those with linkers of at least 10-AA in length induced C-to-T conversions with pTarget plasmids containing spacers of at least 4-bp in length at frequencies that ranged from 1–24% (Fig. 1c and Supplementary Fig. 1a, b). ZFD pairs with a 24-AA linker showed the highest editing efficiencies. To determine the best combinations of linkers, we also measured the editing efficiencies of ZFD pairs that consisted of a left ZFD construct with the 24-AA linker and a right ZFD with variable linkers or vice versa (Fig. 1d and Supplementary Fig. 2). We found that the right ZFD with the 24-AA linker was most active when paired with the left ZFD with the same 24-AA linker. We also found that ZFDs with DddA$_{tox}$ split at G1397 were more efficient than those with DddA$_{tox}$ split at G1333 (Fig. 1c and Supplementary Fig. 1a, b). Cytosines were edited by these most efficient ZFD pairs with high efficiencies of >6.8% in spacer regions of 7–21 bps in length (Fig. 1c and Supplementary Fig. 1a, c).

**Base editing at endogenous target sites**. Next, we investigated whether ZFDs with the 24-AA linker could catalyze C-to-T edits at endogenous chromosomal target sites in human cells. A total of 22 ZFD pairs were designed to target 11 sites (two ZFD pairs per site) in eight genes (Fig. 2 and Supplementary Table 3, 4). Among them, 14 ZFD pairs were constructed from scratch using publically-available zinc finger resources. We also modified previously-characterized ZFNs (specific to *CCR5*[27] and *TRAC*[28]) to make the other eight ZFD pairs. Because ZFN pairs cleave target DNA in spacer regions of 5–7 bps[22] in length, whereas our ZFD pairs preferentially operate in spacer regions of at least 7 bps, we deleted one or two zinc fingers from these ZFN pairs and added a few zinc fingers to make ZFDs (Supplementary Fig. 3). Since the FokI nuclease domain can be fused to either the N- or C-termini of ZFPs to create ZFNs with four different configurations[28], we also constructed two ZFD pairs (*TRAC*-NC in Fig. 2b) with an alternative configuration (shown as NC configuration in Fig. 2a and Supplementary Fig. 4) to test whether split-DddA$_{tox}$ halves can be fused to the N terminus of ZFPs.

In HEK 293 T cells, efficiencies of C-to-T base editing by these ZFDs, including those with the NC configuration, ranged from 1.0% to 60%, whereas indels were rarely induced, showing frequencies of <0.4% (Fig. 2b and Supplementary Fig. 5). As anticipated from our plasmid-based assays described above, two ZFD pairs targeted to the *CCR5* site with a 5-bp spacer were poorly efficient. The other 20 ZFD pairs targeted to sites with spacers of at least 7 bps showed an average editing frequency of 12.0 ± 3.4%, on par with Cas9-derived Base Editor 2 (8.3 ± 2.2%)[29]. In addition to cytosines in the context of T<u>C</u>, those in the context of A<u>C</u> and GC<u>C</u> were also converted to thymine, albeit less efficiently (Fig. 2c–f). Thus, C$_6$ in the context of A<u>C</u> at the *NUMBL* site and C$_7$ in the context of GC<u>C</u> at the *INPP5D*-2 site were converted to T with frequencies of up to 4.6 and 1.9%, respectively.

**Direct delivery of purified ZFD proteins into human cells**. Delivery of purified recombinant gene-editing enzymes, rather than plasmid DNA encoding them, into cells can reduce off-target effects, avoid innate immune responses against foreign

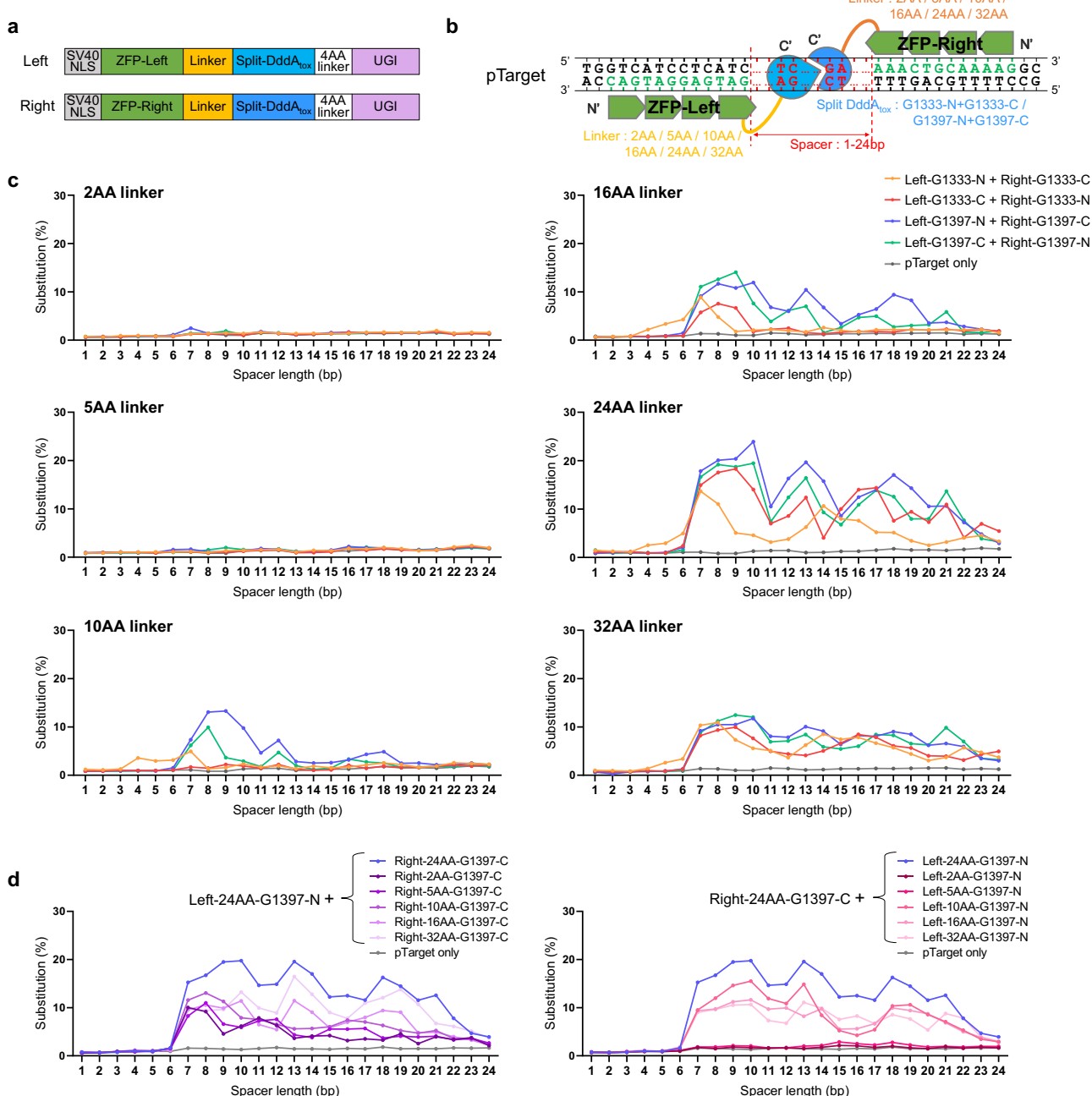

**Fig. 1 Development of ZFDs. a** ZFD (zinc finger deaminase) architecture. Split-DddA$_{tox}$ halves are fused to the C terminus of ZFPs (zinc finger proteins) (C type). **b** Optimization of the ZFD platform using pTarget libraries. pTarget plasmids contain a spacer region that ranges in size from 1–24 bp (shown in red) and ZFP DNA binding sites (shown in green). ZFD constructs contain AA (amino acid) linkers of different lengths (shown in yellow and orange) and different DddA$_{tox}$ split sites and orientations (shown in blue). **c, d** ZFD activities were measured at on-target sites in the pTarget library to examine the effect of the variables described in (**b**). ZFD pairs with linkers of the same (**c**) or different (**d**) lengths in the left and right ZFD were tested. Base editing frequencies were measured by targeted deep sequencing of the relevant region of pTarget plasmids. Data are shown as means from $n = 2$ biologically independent samples. Source data are provided as a Source Data file.

DNA, and exclude the possibility of integration of plasmid DNA fragments into the genome. We and others have shown that ZFPs can spontaneously penetrate into mammalian cells both in vitro[24,25] and in vivo[23]. To demonstrate ZFD protein-mediated base editing in cultured human cells, we chose the highly active ZFD pair targeted to the *TRAC* site (*TRAC*-NC) and purified recombinant *TRAC*-NC proteins, containing one or four copies of a nuclear localization signal (NLS), from *E. coli*. We first assessed the deaminase activity of the ZFD proteins in vitro using a PCR

amplicon containing the *TRAC* site and found that they were highly active, showing efficient DNA cleavage upon treatment with Uracil-Specific Excision Reagent (USER), a mixture of Uracil DNA glycosylase and DNA glycosylase-lyase Endonuclease VIII (Supplementary Fig. 6). We then delivered the *TRAC*-NC ZFD proteins into human leukemia K562 cells via two different methods, electroporation or direct delivery without electroporation. These ZFD proteins were highly efficient, inducing targeted C-to-T conversions at frequencies of up to 27% (electroporation)

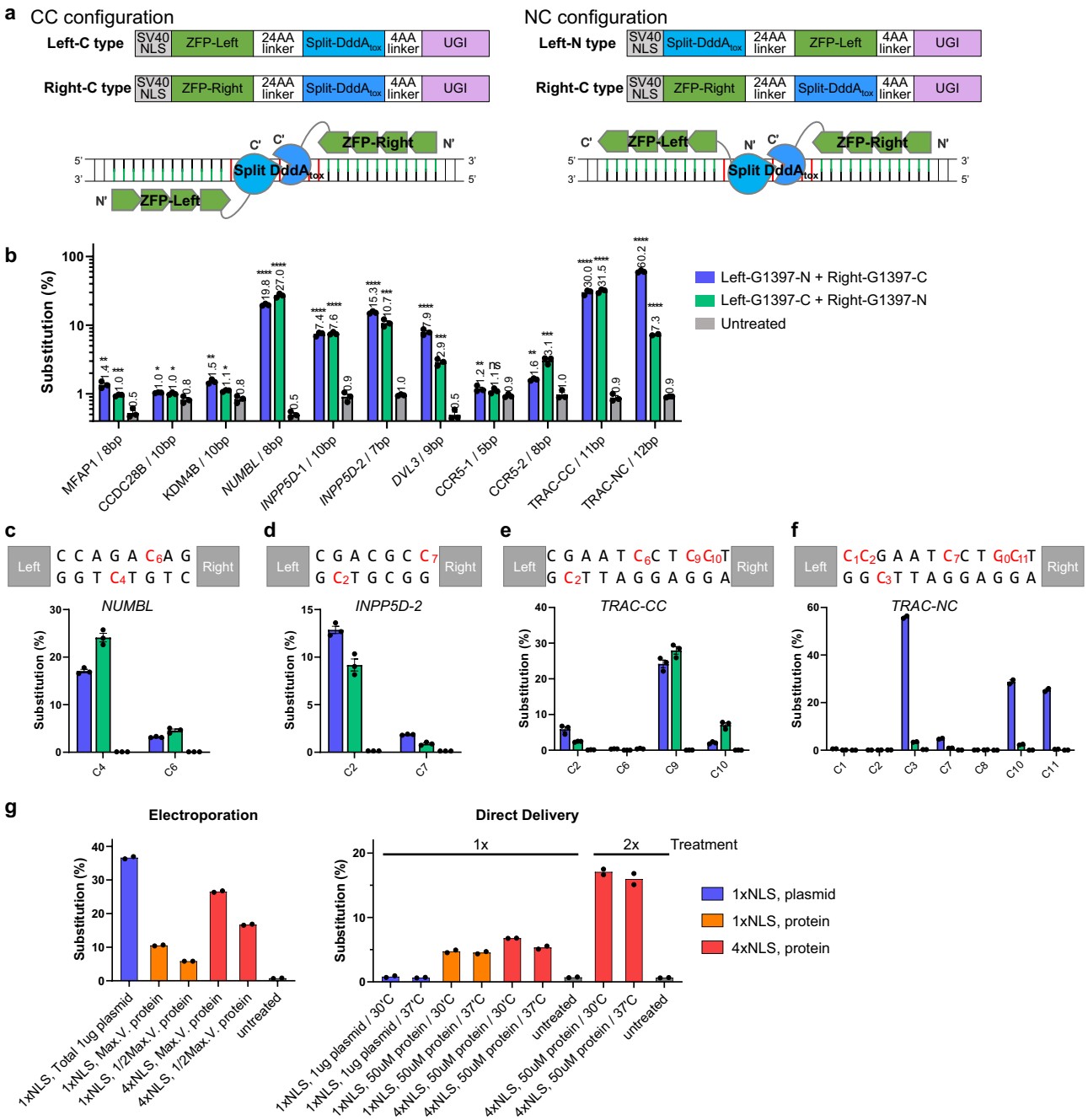

**Fig. 2 Cytosine base editing by ZFDs at endogenous target sites. a** Architecture of nuclear DNA-targeting ZFDs. Split-DddA$_{tox}$ halves are fused to the C terminus (C type) or N terminus (N-type) of ZFPs. ZFD pairs were designed in CC or NC configurations, which are composed of a C-type left ZFD and a C-type right ZFD or an N-type left ZFD and a C-type right ZFD, respectively. **b** Base editing frequencies induced by ZFDs at endogenous target sites in HEK 293 T cells. All statistical analysis for comparing with untreated samples was conducted using unpaired Student's $t$-test (two-tailed) in GraphPad Prism 8. Statistical significance as compared with untreated samples was denoted with *$P \leq 0.05$, **$P \leq 0.01$, ***$P \leq 0.001$, ****$P \leq 0.0001$, n.s. (not significant) $P > 0.05$. Data are shown as means with standard error of the mean (s.e.m.) from $n = 3$ biologically independent samples. **c–f** ZFD-induced base editing efficiencies at each base position within the spacer at the *NUMBL* (**c**), *INPP5D-2* (**d**), *TRAC*-CC (**e**), and *TRAC*-NC (**f**) target sites in HEK 293 T cells. Data are shown as means ± s.e.m. from $n = 3$ biologically independent samples. **g** ZFD-induced base editing frequencies in K562 cells following electroporation or direct delivery of ZFDs or ZFD-encoding plasmids. ZFD proteins with one or four NLSs were tested, and equimolar amounts of left and right ZFDs were used. Electroporation was performed using an Amaxa 4D-Nucleofector. For direct ZFD delivery, K562 cells were incubated with a cell medium containing left and right ZFD proteins. Cells were treated either once (1x) or twice (2x) in the same way. Data are shown as means from $n = 2$ biologically independent samples. Source data are provided as a Source Data file.

and 17% (direct delivery) (Fig. 2g). Taken together, these results show that plasmids encoding ZFDs or purified recombinant ZFD proteins can be used for base editing of nuclear DNA in human cells.

**Mitochondrial DNA editing with ZFDs.** One major advantage of the split-DddA$_{tox}$ system fused to custom-designed DNA-binding proteins is that these programmable deaminases, unlike CRISPR-based systems, can be used for editing organelle DNA

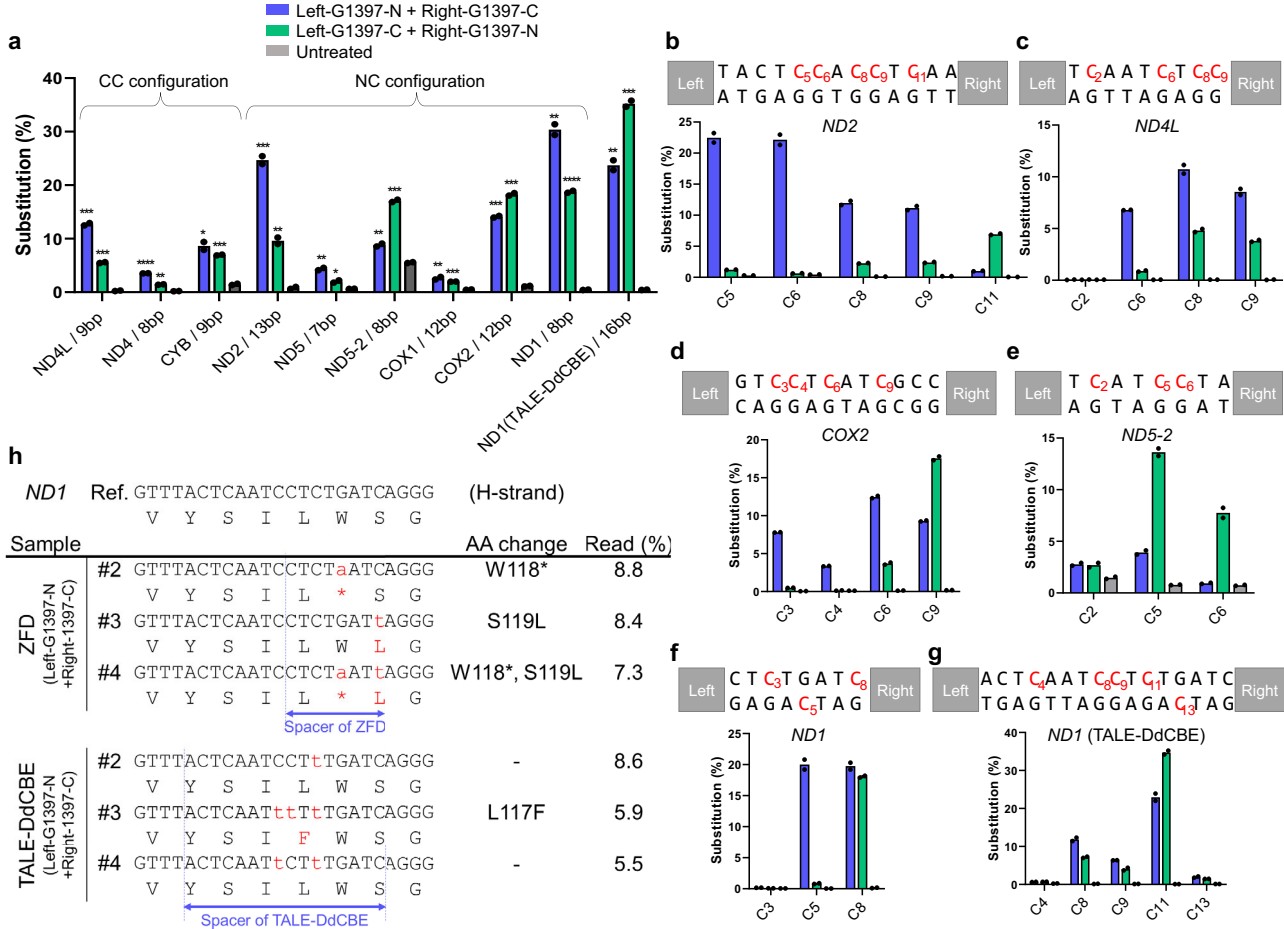

**Fig. 3 mtDNA editing with mitoZFDs. a** Base editing frequencies in mtDNA induced by mitoZFDs and a TALE-DdCBE in HEK 293 T cells. All statistical analysis for comparing with untreated samples was conducted using unpaired Student's $t$-test (two-tailed) in GraphPad Prism 8. Statistical significance as compared with untreated samples was denoted with $*P \leq 0.05$, $**P \leq 0.01$, $***P \leq 0.001$, $****P \leq 0.0001$, n.s. (not significant) $P > 0.05$. Data are shown as means from $n = 2$ biologically independent samples. **b–g** mitoZFD-induced base editing efficiencies at each base position within the spacer at the ND2 (**b**), ND4L (**c**), COX2 (**d**), ND5-2 (**e**), and ND1 (**f**) target sites, and TALE-DdCBE-induced base editing efficiencies at the ND1 (**g**) target site, in HEK 293 T cells. Data are shown as means from $n = 2$ biologically independent samples. **h** Comparison of DNA changes and amino acid changes in the ND1 gene introduced by mitoZFD and TALE-DdCBE. The reference sequence (Ref.) is at the top. In the alleles, the red letters indicate changes in the amino acid sequence. (* indicates a stop codon.) The frequency of sequencing reads (%) for each mutant allele was measured by targeted deep sequencing. The spacer regions for the ZFD pair and the TALE-DdCBE pair are indicated with blue dashed lines. Source data are provided as a Source Data file.

including mtDNA[3] and chloroplast DNA[20,21]. To deliver ZFDs to mitochondria, we constructed mitoZFDs (mitochondria-targeting ZFDs) by linking the mitochondrial targeting signal (MTS) and nuclear export signal (NES) sequences[30] to the N terminus of custom-designed ZFDs targeted to eight mitochondrial genes (Supplementary Fig. 7). ZFP-encoding DNA segments (full sequences are shown in Supplementary Table 5, 6) for these ZFDs were assembled using a publicly available zinc finger resource. These ZFDs were designed to recognize left- and right-binding sites of 12 bps in length separated by a spacer of 7–15 bps. The efficiencies of mtDNA editing in HEK 293 T cells by these mitoZFDs ranged from 2.6–30% ($11 \pm 2\%$ on average, $n = 18$) (Fig. 3a). Interestingly, mitoZFDs with the NC configuration (NC ZFDs) were more efficient ($13 \pm 3\%$, $n = 12$) than those with the CC configuration (CC ZFDs) ($7 \pm 2\%$, $n = 6$). (However, this does not mean that NC ZFDs, in general, are advantageous over CC ZFDs, because different ZFPs are used in NC ZFDs and CC ZFDs. It is possible that CC ZFDs but not NC ZFDs (or vice versa) can be designed to recognize a pre-determined target site.) Most cytosines in the TC context (and those in the TCC context, which can be converted to TTC first and then to TTT) in a spacer region

were converted, with variable efficiencies (Fig. 3b–g). In addition, two cytosines ($C_8$ and $C_9$) in the ND2 site in the context of ACC were edited with fair frequencies of up to 12 and 11%, respectively (Fig. 3b), which suggests that ZFD-mediated C-to-T editing is not limited to the TC motif.

We next isolated single cell-derived clonal populations of mtDNA mutant cells to demonstrate that mitoZFDs are not cytotoxic and that mtDNA mutations are stably maintained after clonal expansion. Among 30 single cell-derived clones obtained from HEK 293 T cells treated with the ND1-specific mitoZFD, five clones contained mutations in the ND1 gene with frequencies ranging from 35–98% (Supplementary Fig. 8a). Likewise, among 36 single cell-derived clones isolated from ND2-mitoZFD-treated cells, we obtained seven base-edited clones that contained ND2 mutations with frequencies that ranged from 26–76% (Supplementary Fig. 8b). All of the other clones had mutations at low frequencies of 0.4–1.0%, most likely resulting from sequencing errors: ZFD-untreated cells had mutations at similar frequencies (Supplementary Fig. 8c). These results show that mitoZFDs induced heteroplasmic mutations unevenly in a population of cells: Thus, most of the ZFD-transfected cells were

homoplasmically wild-type, whereas mutant cells contained heteroplasmic mutations at high frequencies of up to 98%, which were stably maintained after clonal expansion (Supplementary Figs. 9 and 10).

**mitoZFDs and TALE-based DdCBEs.** We found that mutation patterns induced by our *ND1*-specific mitoZFDs were quite different from those induced by two TALE-based DdCBE pairs designed to target the same site (Fig. 3f–h). Thus, the two mitoZFDs catalyzed C-to-T conversions at the $C_5$ or $C_8$ position (Fig. 3f), whereas the DdCBEs induced base conversions at $C_8$, $C_9$, and $C_{11}$ positions (Fig. 3g). As a result, amino-acid changes caused by mitoZFDs were completely different from those caused by DdCBEs (Fig. 3h). Note that our mitoZFDs bind to left- and right-half sites separated by an 8-bp spacer, whereas DdCBEs bind to target sites separated by a 16-bp spacer, which is likely responsible for the differential mutation patterns. These results suggest that mitoZFDs and DdCBEs can be used to generate complementary, diverse mutation patterns in mtDNA.

To further expand possible mutation patterns, we tested whether a ZFD monomer could be combined with a DdCBE monomer to create a functional hybrid pair (Fig. 4a). Ten ZFD/DdCBE hybrid pairs specific to the *ND1* site were highly efficient in HEK 293 T cells, with an average editing frequency of 17 ± 3% (Fig. 4b). In fact, one of the hybrid pairs (TALE-L/ZFD-R1) was more efficient than the two DdCBE pairs and ten ZFD pairs targeted to the same site and exhibited the highest editing frequency (41%) (Fig. 4b). Furthermore, hybrid pairs yielded mutation patterns different from those obtained with ZFDs or DdCBEs (Fig. 4c). Of note, a few hybrid pairs (for example, ZFD-L1/TALE-R and ZFD-L2/TALE-R) induced C-to-T edits at single positions without bystander edits. In contrast, most of the ZFD pairs and DdCBE pairs induced C-to-T conversions at multiple positions in spacer regions. We also found that ZFD/DdCBE hybrid pairs designed to target a Cox2 site were as efficient as ZFD pairs and DdCBE pairs targeted to the same site and produced mutation patterns distinct from those obtained with ZFD pairs and DdCBE pairs (Supplementary Fig. 11). These results demonstrate that ZFD/DdCBE hybrid pairs can create unique mutation patterns and produce certain mutations that cannot be obtained using ZFDs or DdCBEs.

**Mitochondrial genome-wide target specificity of ZFDs.** To investigate whether mitoZFDs exhibit off-target DNA editing in mtDNA, we performed whole mitochondrial genome sequencing using DNA samples isolated from cells transfected with *ND1*- or *ND2*-targeted mitoZFD pairs. Variable amounts (5–500 ng) of mRNA or plasmids encoding these ZFD pairs were transfected into HEK 293 T cells. As expected, on-target editing frequencies were dose-dependent. High doses (100, 200, and 500 ng) of mRNA or plasmids yielded on-target C-to-T edits with frequencies of >30% but also caused hundreds of off-target edits with frequencies of >1.0% in the mitochondrial genome (Supplementary Figs. 12–15). Sequence logos obtained at off-target sites of the ND2-specific mitoZFD showed a preference for the TC context, indicative of the $DddA_{tox}$ substrate specificity (Supplementary Fig. 16). Low doses (5 and 10 ng) of mRNA or plasmids largely avoided these off-target edits but reduced on-target mutation frequencies significantly. A medium dose (50 ng) of mRNA was optimal, avoiding hundreds of off-target edits while maintaining high on-target mutation frequencies. To further eliminate remaining off-target edits, we incorporated R(-5)Q mutations in each zinc finger in the ZFDs to remove non-specific DNA contacts[31]. The resulting ZFD variant pair (shown as QQ in Fig. 5) retained high on-target activity and showed exquisite

specificity with almost no off-target edits, compared to untreated mtDNA. In particular, the specificity ratio was improved by 8.2-fold, when 50 ng mRNA of mitoZFD with R(-5)Q mutations was used, compared to 200 ng of WT mitoZFD mRNA (Fig. 5e). Additionally, we assessed off-target editing in nuclear DNA at sites with high sequence homology to ZFP-binding sequences. No off-target edits were detectably induced by the *ND4L* mitoZFD at three homologous sites that differ from the on-target site by a single or two nucleotides (Supplementary Fig. 17a), whereas off-target edits were induced with low (~1.0%) frequencies by the *ND2* mitoZFD at a homologous site with a one-nucleotide mismatch (Supplementary Fig. 17b). Use of the QQ variant pair reduced off-target edit frequency to 0.4% at this site (Supplementary Fig. 17b).

## Discussion
Base editing is a relatively new method for generating targeted nucleotide substitutions without double-strand DNA cleavage or a repair DNA template. Base editing enables C-to-T or A-to-G conversions in cell lines[3,7,8], animals[19,32,33], and plants[20,21,34], allowing researchers to study the functional effects of single-nucleotide polymorphisms (SNPs) and to correct disease-causing point mutations for therapeutic applications. Two types of base editors have been developed: CRISPR-derived ABEs and CBEs and DddA-derived CBEs. CRISPR-derived base editors are composed of catalytically-impaired Cas9 or Cas12a variants as DNA-binding units and single-strand DNA-specific deaminases originated from rat, sea lamprey, or *E. coli*[7–9,35], whereas DdCBEs are composed of TALE DNA-binding arrays and double-strand DNA-specific $DddA_{tox}$[3]. In this study, we presented an alternative type of base editors, ZFDs, composed of zinc finger DNA-binding arrays and $DddA_{tox}$.

Compared to DdCBEs, ZFDs are smaller in size, because the ZFPs in ZFDs are compact, whereas the TALE arrays in DdCBEs are bulky. As a result, a ZFD pair-encoding gene but not a DdCBE pair-encoding gene can be readily packaged in an AAV vector with a small cargo space. In addition, compact ZFPs are engineering friendly, making it possible to fuse split-$DddA_{tox}$ halves to either the C or N terminus of a ZFP, creating ZFDs that operate either upstream or downstream of a ZFP-binding site. Furthermore, recombinant ZFD proteins can penetrate into human cells spontaneously without electroporation or lipofection, potentially enabling gene-free gene therapy. Last but not least, ZFD pairs or ZFD/DdCBE hybrid pairs can produce unique mutation patterns, which cannot be obtained using DdCBEs alone. These properties will make ZFDs a powerful platform for modeling and treating mitochondrial diseases.

We expect that ZFDs can be further engineered to improve efficiency and specificity. Here, we showed that use of ZFP variants and ZFD mRNA can reduce off-target activity. $DddA_{tox}$ can also be engineered to avoid ZFD off-target mutations. ZFDs with enhanced efficiency and precision could pave the way for correcting pathogenic mitochondrial DNA mutations in human embryos, fetuses, and patients.

## Methods
**Plasmid construction.** p3s-ZFD plasmids for mammalian expression were created by modifying the p3s-ABE7.10 plasmid (addgene, #113128)[32] after digestion with HindIII and XhoI (NEB). The digested p3s plasmid and synthesized insert DNAs were assembled using a HiFi DNA assembly kit (NEB). All insert DNAs, which encoded MTS, ZFP (from Toolgen[36], Sangamo[28], and Barbas module[37]), split-DddA, or UGI, were synthesized by IDT.

The pTarget plasmids were designed for determining the optimal length of the spacer sequence for ZFD activity. Each pTarget plasmid, which contains two ZFP-binding sites with a spacer of variable length between them, was constructed by inserting the ZFP-binding sequences and a spacer sequence into the pRGS-CCR5-

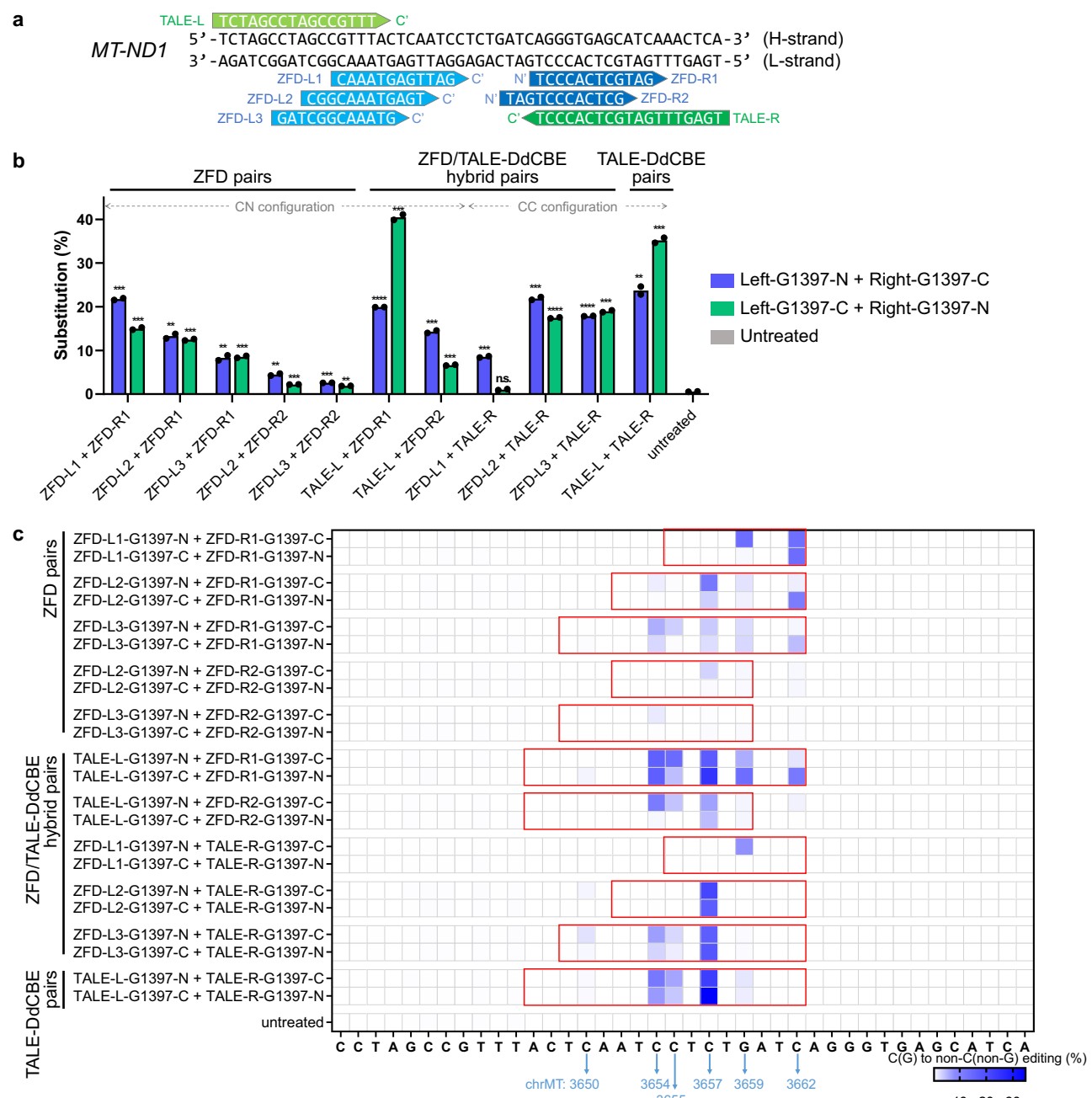

**Fig. 4 Activity of mitoZFDs, TALE-based DdCBEs, and ZFD/DdCBE hybrid pairs. a** DNA sequences of the binding regions of the mitoZFD and TALE-DdCBE pairs. Sites recognized by the TALE-DdCBEs are highlighted in green and for the mitoZFDs in blue. The upper sequence represents the mtDNA heavy strand and the lower sequence represents the mtDNA light strand. **b** Frequencies of cytosines edited by ZFDs, TALE-DdCBEs, and ZFD/DdCBE hybrid pairs. All statistical analysis for comparing with untreated samples was conducted using unpaired Student's $t$-test (two-tailed) in GraphPad Prism 8. Statistical significance as compared with untreated samples was denoted with *$P \leq 0.05$, **$P \leq 0.01$, ***$P \leq 0.001$, ****$P \leq 0.0001$, n.s. (not significant) $P > 0.05$. Data are shown as means from $n = 2$ biologically independent samples. **c** Heat maps of base editing activities at each base position. The red box indicates the spacer region for each construct. The blue arrows indicate the position in the mtDNA. Source data are provided as a Source Data file.

NHEJ reporter plasmid after it had been digested with two enzymes (EcoRI and BamHI, NEB), which recognize sites between the RFP and EGFP sequences.

pET-ZFD plasmids for protein production in *E. coli* were created by modifying the pET-Hisx6-rAPOBEC1-XTEN-nCas9-UGI-NLS plasmid (addgene, #89508)[33] after digestion with NcoI and XhoI (NEB). ZFD sequences were amplified from the p3s-ZFD plasmid using PCR, and Hisx6 tag and GST tag sequences were synthesized as oligonucleotides (Macrogen). All plasmids were generated using a HiFi DNA assembly kit (NEB) to insert sequences encoding the ZFD and tag for protein purification into the digested pET plasmid.

DH5α chemically competent *E. coli* cells were used for transformation of plasmids, and plasmids were purified with an AccuPrep Plasmid Mini Extraction

Kit (Bioneer) according to the manufacturer's protocol. The desired plasmids were selected after confirming the entire sequence with Sanger sequencing.

**ZFD design scheme**. In this paper, ZFDs were prepared in two ways. First, ZFDs were made by removing or attaching zinc finger protein from previously constructed ZFN. Unlike the ZFN whose optimal spacer is 5~6 bp, the optimal spacer of ZFD is ≥7 bp, so the zinc finger on the back part was removed to widen the spacer. When manufacturing zinc finger array protein, it is recommended to use four or more fingers. If there were <4 after removing one zinc finger, we added one zinc finger to the front part (Supplementary Fig. 3). Second, ZFDs were de novo

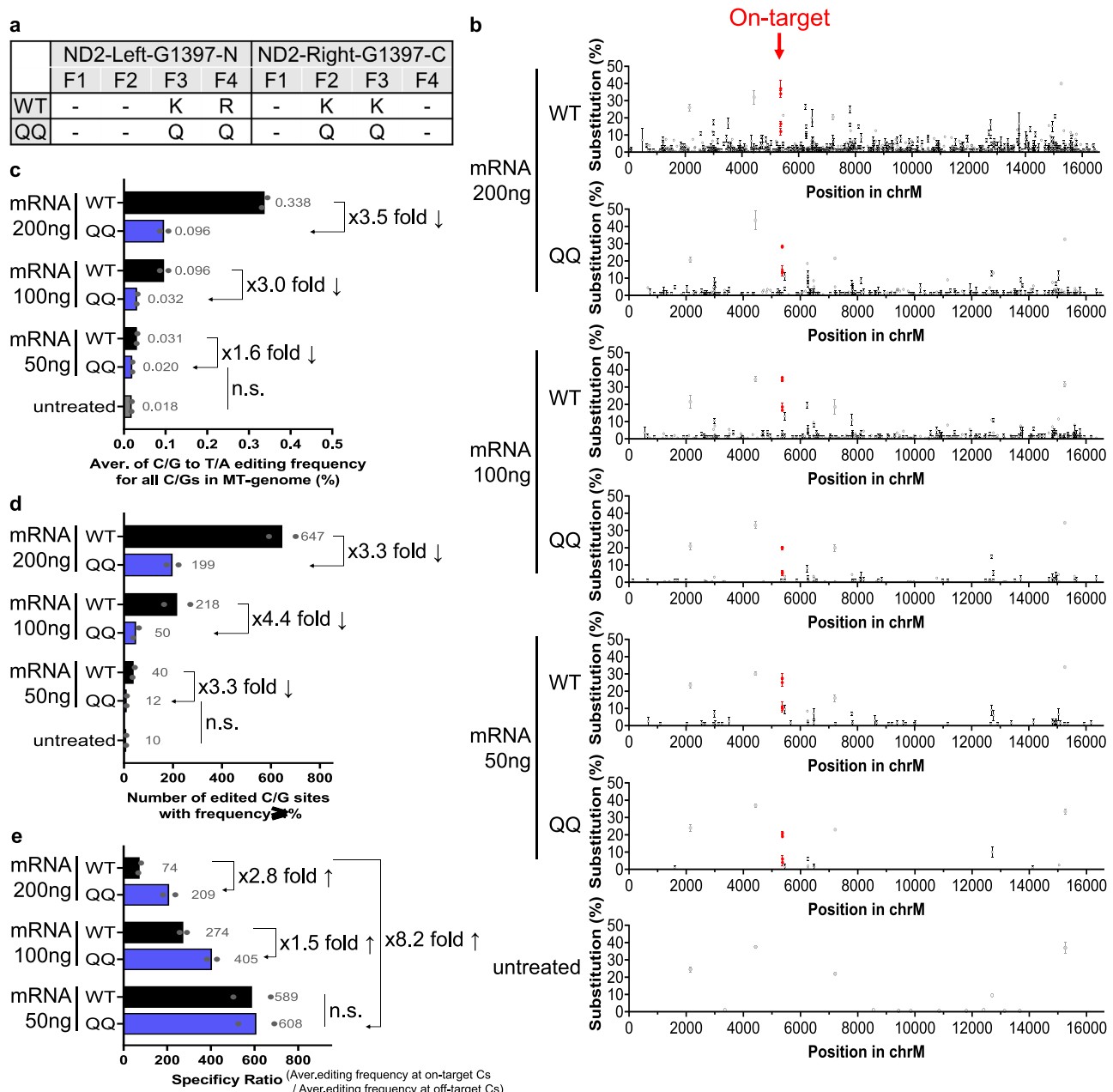

**Fig. 5 Improving the mitochondrial genome-wide target specificity of mitoZFDs. a** QQ mitoZFD variants contain R(-5)Q mutations in each zinc finger in the ZFD to remove non-specific DNA contacts. F1–F4; Finger 1–4. (If no R was present at position-5 of the zinc finger framework, a nearby K or R was converted to Q.) **b** Whole-mtDNA sequencing of mitoZFD-treated cells. Editing frequencies at on-target and off-target sites are indicated by red and black dots, respectively. Error bars are shown as standard error of the mean (s.e.m.) for $n = 2$ biologically independent samples. All C/G-to-T/A base changes present at frequencies >1% are presented. **c–e** Editing efficiencies and specificities depend on the dose of ZFD-encoding mRNA delivered. Data are shown as means from $n = 2$ biologically independent samples. **c** The average C/G-to-T/A editing frequency for all C/Gs in the mitochondrial genome. **d** The number of edited C/G sites with base editing frequencies >1%. **e** The specificity ratio was calculated by dividing (average editing frequency at on-target Cs) by (average editing frequency at off-target Cs). Source data are provided as a Source Data file.

assembled using a publicly available zinc finger resource (from Toolgen[36]). In this resource, 33 ZFP were recommended for use. In this way, since the binding ability of the ZFP array is considered critical for ZFD activity, choose the N-type or C type that can use the recommended ZFP at the DNA binding site. Finally, the generated ZFP was cloned using different original plasmids depending on the nucleus and mitochondria targets (Supplementary Figs. 4 and 7).

**HEK 293 T cell culture and transfection**. HEK 293 T cells (ATCC, CRL-11268) were cultured in Dulbecco's Modified Eagle Medium (Welgene) supplemented with 10% fetal bovine serum (Welgene) and 1% antibiotic-antimycotic solution (Welgene). HEK 293 T cells ($7.5 \times 10^4$) were seeded into 48-well plates. After 18–24 h,

cells were transfected at 70–80% confluency with plasmids encoding left ZFD and right ZFD (500 ng each, for the full dose), or together with a pTarget plasmid (10 ng), using Lipofectamine 2000 (1.5 μL, Invitrogen). Cells were harvested at 96 h post treatment, after which they were lysed by incubation at 55 °C for 1 h, and then at 95 °C for 10 min, in 100 μL of cell lysis buffer (50 mM Tris-HCl, pH 8.0 (Sigma-Aldrich), 1 mM EDTA (Sigma-Aldrich), 0.005% sodium dodecyl sulfate (Sigma-Aldrich)) supplemented with 5 μL of Proteinase K (Qiagen).

For whole-mtDNA sequencing, HEK 293 T cells were transfected with serially diluted concentrations of plasmid or mRNA encoding *ND1*- or *ND2*-targeted mitoZFD pairs. Because more cells were required for mtDNA extraction than for analysis of the editing efficiency at defined sites, four samples, transfected under the same conditions as described above, were collected as a single sample (at four times

the scale). In the manuscript, the amounts of constructs (ng) that were delivered per $7.5 \times 10^4$ cells are indicated. mtDNA was isolated from cells at 96 h post transfection.

**K562 cell culture and transfection**. K562 cells (ATCC, CCL-243) were maintained in RPMI 1640 supplemented with 10% fetal bovine serum (Welgene) and 1% antibiotic–antimycotic solution (Welgene).

For ZFD delivery into K562 cells by electroporation, an Amaxa 4D–Nucleofector™ X Unit system with program FF-120 (Lonza) was used. The maximum volume of substrate solution added to each sample was 2 µL when using a 16-well Nucleocuvette™ Strip. 220 pmol (for maximum capacity) or 110 pmol (for half of the maximum capacity) of each of the left and right ZFD proteins, or 500 ng of plasmid encoding left and right ZFD, were transfected into K562 cells ($1 \times 10^5$). At 96 h post treatment, cells were collected by centrifugation at 100 g for 5 min, and lysed by incubation at 55 °C for 1 h, and then at 95 °C for 10 min, in 100 µL of cell lysis buffer (50 mM Tris-HCl, pH 8.0 (Sigma-Aldrich), 1 mM EDTA (Sigma-Aldrich), 0.005% sodium dodecyl sulfate (Sigma-Aldrich)) supplemented with 5 µL of Proteinase K (Qiagen).

For direct delivery of ZFD or ZFD-encoding plasmids into K562 cells, we referred to a method previously used for direct delivery of ZFN[24]. A mixture of left and right ZFD proteins (at a final concentration of 50 µM) or a mixture of plasmids encoding left and right ZFD (500 ng each) was diluted into serum-free medium containing 100 mM L-arginine and 90 µM ZnCl₂ at pH 7.4 to a final volume of 20 µL. K562 cells ($1 \times 10^5$) were centrifuged at 100 g for 5 min, and the supernatant was removed. The cells were then resuspended in the diluted ZFD solution and incubated for 1 h at 37 °C. After incubation, cells were centrifuged at 100 g for 5 min, and then resuspended in fresh culture medium. Cells were maintained at 30 °C (for a transient hypothermic condition) or 37 °C for 18 h, and then for two more days at 37 °C. Some cells were subjected to a second treatment, following the above process. Cells were analyzed 96 h after treatment.

**ZFD protein expression and purification**. The plasmids encoding each pair of ZFDs (Supplementary Table 7), each with a C-terminal GST tag, were transformed into Rosetta (DE3) competent cells, which were then cultured on LB-agar plates containing 50 µg/ml kanamycin. After incubation overnight, a single colony was picked and grown overnight (pre-culture) in LB broth containing 50 µg/ml kanamycin and 100 µM ZnCl₂ at 37 °C. The next day, part of the pre-culture was transferred to a large volume of LB broth, which was incubated at 37 °C with shaking at 200 rpm until the absorbance, A600 nm = ~0.5–0.70. The cultures were put on ice for about 1 h, after which ZFD protein expression was induced by the addition of 0.5 mM of isopropyl β-D-1-thiogalactopyranoside (IPTG; GoldBio) and the culture was incubated at 18 °C for 14–16 h.

Protein purification steps were carried out at 0–4 °C. For cell lysis, the cells were harvested by centrifugation at 5000 g for 10 min and then resuspended in lysis buffer (50 mM Tris-HCl (Sigma-Aldrich), 500 mM NaCl (Sigma-Aldrich), 1 mM MgCl₂ (Sigma-Aldrich), 10 mM 1,4-dithiothreitol (DTT; GoldBio), 1% Triton X-100 (Sigma-Aldrich), 10% glycerol, 1 mM phenylmethylsulfonyl fluoride (Sigma-Aldrich), 1 mg/ml lysozyme from chicken egg white (Sigma-Aldrich), 100 µM ZnCl₂ (Sigma-Aldrich), 100 mM arginine (Sigma-Aldrich), pH 8.0). For further lysis, cells were sonicated (3 min total, 5 s on, 10 s off), after which the solution was centrifuged at 18,500 g to clear the lysate. The supernatant was then incubated with Glutathione Sepharose 4B (GE healthcare) for 1 h with gentle rotation. After this incubation, the resin-lysate mixture was loaded onto a column, which was then washed three times with wash buffer (50 mM Tris-HCl (Sigma-Aldrich), 500 mM NaCl (Sigma-Aldrich), 10 mM DTT (GoldBio), 1 mM MgCl₂ (Sigma-Aldrich), 100 µM ZnCl₂ (Sigma-Aldrich), 10% glycerol, 100 mM arginine (Sigma-Aldrich), pH 8.0). The bound proteins were eluted with elution buffer (50 mM Tris-HCl (Sigma-Aldrich), 500 mM NaCl (Sigma-Aldrich), 1 mM MgCl₂ (Sigma-Aldrich), 40 mM glutathione (Sigma-Aldrich), 10% glycerol, 1 mM DTT (GoldBio), 100 µM ZnCl₂ (Sigma-Aldrich), 100 mM arginine (Sigma-Aldrich), pH 8.0). Finally, the eluted proteins were concentrated to a concentration of ~15 ng/µL (200–240 pmol/µL, depending on the protein size) using an Amicon Ultra-4 column with a 30,000 kDa cutoff (Millipore) at 5,000 g.

**In vitro deamination of PCR amplicons by ZFD**. An amplicon containing the *TRAC* site (Supplementary Table 8) was prepared using PCR. Eight microgram of the amplicon was incubated with 2 µg of each ZFD protein (Left-G1397-N and Right-G1397-C) in NEB3.1 buffer containing 100 µM ZnCl₂ for 1–2 h at 37 °C. Following the reaction, ZFD proteins were removed by incubating with 4 µL of Proteinase K solution (Qiagen) for 30 min at 55 °C, and the amplicon was purified using a PCR purification kit (MGmed). One microgram of the purified amplicon was incubated with 2 units of USER enzyme (NEB) for 1 h at 37 °C. Then, the amplicon was incubated with 4 µL of Proteinase K solution (Qiagen) and purified again using a PCR purification kit (MGmed). The product was subjected to electrophoresis on an agarose gel and imaged.

**mRNA preparation**. DNA templates containing a T7 RNA polymerase promoter upstream of the ZFD sequence were generated from p3s-ZFD plasmids by PCR amplification using Q5 high fidelity DNA polymerase (NEB) with forward and reverse primers (Forward: 5'-CATCAATGGGCGTGGATAG-3', Reverse: 5'-GACACCTACTCAGACAATGC-3'). mRNAs were synthesized in vitro using an mMESSAGE mMACHINE™ T7 ULTRA Transcription Kit (Thermo Fisher). In vitro transcribed mRNAs were purified using a MEGAclear™ Transcription Clean-Up Kit (Thermo Fisher) according to the manufacturer's protocol.

**Targeted deep sequencing**. Nested PCR was used to produce libraries for next-generation sequencing (NGS). The region of interest was first amplified by PCR using KAPA HiFi HotStart PCR polymerase (Roche). To generate NGS libraries, amplicons were amplified again using TruSeq DNA-RNA CD index-containing primers to label each fragment with adapter and index sequences. Final PCR products were purified using a PCR purification kit (MGmed) and sequenced using a MiniSeq sequencer (Illumina) with a GenerateFASTQ workflow. Primer sequences for targeted deep sequencing are listed in Supplementary Table 10. Substitution and indel frequencies from targeted deep sequencing data were calculated with source code (https://github.com/ibs-cge/maund, written by BotBot Inc.).

**Whole mitochondrial genome sequencing**. For whole mitochondrial genome sequencing, three steps were required: mtDNA extraction from isolated mitochondria, NGS library generation, and NGS. First, $3 \times 10^5$ HEK 293 T cells were trypsinized and collected by centrifugation (500 g, 4 min, 4 °C) 96 h after transfection with *ND1*- or *ND2*-targeted mitoZFD pairs. Then, cells were washed with ice-cold phosphate-buffered saline (Welgene), and collected again by centrifugation. The supernatant was removed, and the mitochondria were isolated from cultured cells using the reagent-based method of the Mitochondria Isolation Kit for Cultured Cells (Thermo Fisher) according to the manufacturer's protocol. mtDNA was extracted from isolated mitochondria with a DNeasy Blood & Tissue Kit (Qiagen). To generate an NGS library from the extracted mtDNA, we used an Illumina DNA Prep kit with Nextera™ DNA CD Indexes (Illumina). Finally, the libraries were pooled and loaded onto a MiniSeq sequencer (Illumina). The average sequencing depth was >50.

**Analysis of mitochondrial genome-wide DNA editing**. To analyze NGS data from whole mitochondrial genome sequencing, we referred to a method previously used for DNA off-target analysis of TALE-DdCBE[3]. First, we aligned the Fastq files to the GRCh38.p13 (release v102) reference genome using BWA (v.0.7.17), and generated BAM files with SAMtools (v.1.9) by fixing read pairing information and flags. Then, we used the REDItoolDenovo.py script from REDItools (v.1.2.1)[38] to identify, among all cytosine and guanines in the mitochondrial genome, the positions with conversion rates ≥1%. We excluded positions with conversion rates ≥50% in all samples, regarding these as single-nucleotide variations in the cell lines. We also excluded the on-target sites for each ZFD treatment. We considered the remaining positions to be off-target sites and counted the number of edited C/G nucleotides with an editing frequency ≥1%. We averaged the conversion rates at each base position in the off-target sites to calculate the average C/G to T/A editing frequency for all C/Gs in the mitochondrial genome. Specificity ratios were calculated by dividing the average on-target editing frequency by the average off-target editing frequency. Mitochondrial genome-wide graphs were created by plotting the conversion rates at on-target and off-target sites.

**Data visualization**. GraphPad Prism 8, Adobe Illustrator CS6, Microsoft Excel 2016, and PowerPoint 2016 were used for generating figures and tables.

**Reporting summary**. Further information on research design is available in the Nature Research Reporting Summary linked to this article.

## Data availability

DNA sequencing data have been deposited in the National Center for Biotechnology Information(NCBI) Sequence Read Archive (SRA) database with BioProject accession code PRJNA756903. The data underlying Figs. 1–5 and Supplementary figs. 1, 2, 5, 6, 11, 12, 13, 14, 15, and 17 in this study are provided as a Source Data file. DNA sequences of target sites and amino acid sequences of ZFDs are provided in the Supplementary Table. Source data are provided with this paper. The plasmid encoding each ZFD pair for TRAC-NC site in nuclear DNA and ND1 and ND2 site in mitochondrial DNA are available from Addgene (TRAC-NC ZFD pair; Addgene #180772 and #180773, ND1 mitoZFD pair; Addgene #180768 and #180769, ND2 mitoZFD pair; Addgene #180770 and #180771). Any other additional relevant data are available from the authors upon request.

## Code availability

Base editing frequencies and indel frequencies from targeted deep sequencing data were calculated with source code (https://github.com/ibs-cge/maund, written by BotBot Inc.).

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

## Acknowledgements

This work was supported by the Institute for Basic Science (IBS-R021-D1 to J.-S.K.).

## Author contributions

J.-S.K. supervised the research. K.L., S.-I.C., and J.-S.K. designed the study. K.L. and S.-I.C. performed the experiments and carried out bioinformatics analysis. All authors discussed the results and wrote the manuscript. Correspondence and requests for materials should be addressed to J.-S.K (jskim01@snu.ac.kr).

## Competing interests

J.-S.K. is a founder of and shareholder in ToolGen, Inc. The other authors declare no competing interests.
