## [Peer Review File · Nature Communications]

Reviewers' Comments:

Reviewer #1:

Remarks to the Author:

In this manuscript, Lim, Cho and Kim developed the zinc finger deaminase (ZFD) tool as a new version of base editing technology that can change a single base in the nuclear and mitochondrial genomes without cutting the DNA. The previous study by Mok et al has established the concept by using split forms of DddA cytosine deaminase fused to TALE. Unlike the original base editors that are based on the CRISPR-Cas system and require guide RNA for targeting, protein-based tools can also be applied for organelle genome editing. To develop ZFD with practically allowable efficiency, the authors have tested a series of configurations. Although the target design is rather complex and efficiency is less robust for ZF-based tools in general, one of the advantages is its smaller size. As demonstrated, preparation of the recombinant protein is also possible, allowing DNA/RNA-free base editing. These features may be attractive for applications such as gene therapy. This reviewer feels that the manuscript is well-written, consistent, and worthy of publication, although there are a few points that need to be addressed, as described below.

The comparison and hybrid with TALE has been done on only one site. To generalize these results, at least two sites are needed.

L167-171 may be difficult for the average reader to understand. Does this intend to mean "mitoZFDs induced heteroplasmic mutations unevenly in a population of cells" ?

As a user guide, it would be helpful to provide the target design scheme for each ZFD version, ideally with illustrations.

Reviewer #2:

Remarks to the Author:

To authors,

In this work, Lim et al. combined split DddAtox and custom-designed zinc finger proteins (ZFPs) to generate a new base editing tool: zinc finger deaminase (ZFD). To this end, they optimized architecture to develop ZFD and achieved efficient base editing on both nuclear and mitochondrial DNA. Overall, although the strategy is similar to that described by Mok et al. the authors did create a new base editing tool. They also compared the ZFD with other methods and found that, CRISPR-based technologies can rapidly and precisely edit nuclear DNA, but not mitochondrial DNA, because they rely on a guide RNA to target the genome. Therefore, ZFD has the advantages in modeling mitochondrial diseases and exploring fundamental questions. Meanwhile, compared with TALE-based DdCBEs, which also achieved efficient base editing on mitochondrial DNA, ZFD is smaller and suitable for AAV package and is more friendly for engineering, which is helpful for gene therapy. In the terms of editing outcomes, ZFN or ZFD/DdCBE hybrid can also produce unique mutation patterns. Together, I believe the paper is appropriate for publication in Nature Communications. Nevertheless, some concerns need to be well addressed before publication.

Comments/Suggestions:

1. In Fig. 2b, the editing efficiency of TRAC-NC was far higher than that of TRAC-CC. Does this mean that ZFD with NC configuration has more advantages than ZFD with CC configuration? The authors should test more and show more information about ZFD with NC configuration.
2. The authors showed the editing efficiency and statistical significance up the bars in Fig. 2b, but not the other similar figures (like Fig. 3a, Fig. 4b...). Why? In addition, figure legends should provide information about the statistical method used.
3. I would like to see the authors show the probability sequence logo of the region flanking mutated cytosines.
4. To make the results more convincing, the authors should also show ZFD off-target DNA editing in nuclear DNA and give specific statistical numbers, such as average fold change of both on-target and off-target for ZFD together with QQ (ZFD variant).
5. The Split-DddAtox orientation of the ZFD architecture should be labeled, which may be more helpful for understanding "Left-G1333-N + Right-G1333-C" or "Left-G1333-C + Right-G1333-N".
6. To make ZFD, the authors deleted one or two zinc fingers and added a few zinc fingers. Please

provide more detailed information or references.

7. More information on efficiency and off-target effects on the nuclear DNA or mitochondrial DNA is helpful for the discussion.

Point-by-point response

We would like to thank the two anonymous reviewers for their helpful comments. We addressed various issues raised by the reviewers, as shown below, and highlighted textual changes in our revised manuscript for tracking.

Reviewer #1 (Remarks to the Author):

In this manuscript, Lim, Cho and Kim developed the zinc finger deaminase (ZFD) tool as a new version of base editing technology that can change a single base in the nuclear and mitochondrial genomes without cutting the DNA. The previous study by Mok et al has established the concept by using split forms of DddA cytosine deaminase fused to TALE. Unlike the original base editors that are based on the CRISPR-Cas system and require guide RNA for targeting, protein-based tools can also be applied for organelle genome editing. To develop ZFD with practically allowable efficiency, the authors have tested a series of configurations. Although the target design is rather complex and efficiency is less robust for ZF-based tools in general, one of the advantages is its smaller size. As demonstrated, preparation of the recombinant protein is also possible, allowing DNA/RNA-free base editing. These features may be attractive for applications such as gene therapy. This reviewer feels that the manuscript is well-written, consistent, and worthy of publication, although there are a few points that need to be addressed, as described below.

The comparison and hybrid with TALE has been done on only one site. To generalize these results, at least two sites are needed.

Response: We have measured base editing activities of new ZFD/DdCBE hybrid pairs at a Cox2 site and added the data in Supplementary Fig.11. We have described the result on p. 10, lines 192-195, as follows: “We also found that ZFD/DdCBE hybrid pairs designed to target a Cox2 site were as efficient as ZFD pairs and DdCBE pairs targeted to the same site and produced mutation patterns distinct from those obtained with ZFD pairs and DdCBE pairs (Supplementary Fig. 11).”

Supplementary Fig. 11. Base editing activities of mitoZFDs, TALE-based DdCBEs, and ZFD/DdCBE hybrid pairs at the Cox2 site.

L167-171 may be difficult for the average reader to understand. Does this intend to mean “mitoZFDs induced heteroplasmic mutations unevenly in a population of cells” ?

Response: This reviewer is right! We have now made appropriate changes in the sentence to be more clear, as follows: “These results show that mitoZFDs induced heteroplasmic mutations unevenly in a population of cells: Thus, most of the ZFD-transfected cells were homoplasmically wild-type, ...”

As a user guide, it would be helpful to provide the target design scheme for each ZFD version, ideally with illustrations.

Response: Illustrations of ZFD constructs are shown in Supplementary Fig. 3, Supplementary Fig. 4 (nuclear DNA editing) and Supplementary Fig. 7 (mitochondrial DNA editing). We have added the following paragraph in the Methods, as shown below:

ZFD design scheme

ZFPs used for the construction of ZFDs were prepared in two ways: 1) reengineering of ZFPs in pre-characterized ZFNs, and 2) de novo assembly. In the first approach, new ZFPs were made by removing or adding zinc fingers in pre-characterized ZFPs (Supplementary Fig. 3.) to make ZFDs that operate on a spacer of more than 7 bp in length. In the second approach, ZFPs were assembled from scratch using a publicly available zinc finger resource³⁶. In this resource, 33 zinc fingers were recommended for use. Sequences encoding ZFPs were fused to those encoding split DddA_{tox} to produce ZFDs from expression plasmids (Supplementary Fig. 4 and 7).

Supplementary Figure 3.

Supplementary figure 3. Designing the ZFD pairs by modifying previously-characterized ZFNs.

Supplementary Figure 4

Supplementary figure 4. Possible ZFD architectures and DddA_{tox} orientations.

Supplementary Figure 7

a CC configuration

c CN configuration

b NC configuration

d NN configuration

Supplementary figure 7.

Possible mitoZFD architectures.

Reviewer #2 (Remarks to the Author):

To authors,

In this work, Lim et al. combined split DddAtox and custom-designed zinc finger proteins (ZFPs) to generate a new base editing tool: zinc finger deaminase (ZFD). To this end, they optimized architecture to develop ZFD and achieved efficient base editing on both nuclear and mitochondrial DNA. Overall, although the strategy is similar to that described by Mok et al. the authors did create a new base editing tool. They also compared the ZFD with other methods and found that, CRISPR-based technologies can rapidly and precisely edit nuclear DNA, but not mitochondrial DNA, because they rely on a guide RNA to target the genome. Therefore, ZFD has the advantages in modeling mitochondrial diseases and exploring fundamental questions. Meanwhile, compared with TALE-based DdCBEs, which also achieved efficient base editing on mitochondrial DNA, ZFD is smaller and suitable for AAV package and is more friendly for engineering, which is helpful for gene therapy. In the terms of editing outcomes, ZFN or ZFD/DdCBE hybrid can also produce unique mutation patterns. Together, I believe the paper is appropriate for publication in Nature Communications. Nevertheless, some concerns need to be well addressed before publication.

Comments/Suggestions:

1. In Fig. 2b, the editing efficiency of TRAC-NC was far higher than that of TRAC-CC. Does this mean that ZFD with NC configuration has more advantages than ZFD with CC configuration? The authors should test more and show more information about ZFD with NC configuration.

Response: In fact, we tested 12 additional ZFDs with the NC configuration (NC ZFDs) and compared their base editing activities with those with the CC configuration (CC ZFDs). Interestingly, NC ZFDs were more efficient ($13\pm 3\%$, $n = 12$) than CC ZFDs ($7\pm 2\%$, $n = 6$) (Fig. 3a). However, this does not mean that NC ZFDs, in general, are advantageous over CC ZFDs, because different ZFPs are used in NC ZFDs and CC ZFDs. It is possible that CC ZFDs but not NC ZFDs (or vice versa) can be designed to recognize a pre-determined target site. We have now added this comment in the text on p.8, lines 147-152.

2. The authors showed the editing efficiency and statistical significance up the bars in Fig. 2b, but not the other similar figures (like Fig. 3a, Fig. 4b...). Why? In addition, figure legends should provide information about the statistical method used.

Response: We have now shown statistical significance in other figures and added the following sentences in the figure legends: "All statistical analysis was conducted using unpaired t-test (two-tailed) in GraphPad Prism 8. Statistical significance as compared with untreated samples was denoted with * = $P\leq 0.05$, ** = $P\leq 0.01$, *** = $P\leq 0.001$, **** = $P\leq 0.0001$, n.s. (not significant) = $P> 0.05$. Data are shown as means with standard error of the mean (s.e.m.) from $n=2$ or 3 biologically independent samples."

3. I would like to see the authors show the probability sequence logo of the region flanking mutated cytosines.

Response: We have added Supplementary Fig. 16 and described the result in the main text on p. 10, lines 207-209, as follows: “Sequence logos obtained at off-target sites of the ND2-specific mitoZFD showed a preference for the TC context, indicative

of the DddA_{tox} substrate specificity (Supplementary Fig. 16).”

Supplementary figure 16. Sequence logos at off-target sites of ND2-targeted mitoZFD.

4. To make the results more convincing, the authors should also show ZFD off-target DNA editing in nuclear DNA and give specific statistical numbers, such as average fold change of both on-target and off-target for ZFD together with QQ (ZFD variant).

Response: We have added Supplementary Fig. 17 and described the results in the main text on p. 11, lines 215-224, as follows: “Additionally, we assessed off-target

editing in nuclear DNA at sites with high sequence homology to ZFP-binding sequences. No off-target edits were detectably induced by the *ND4L* mitoZFD at three homologous sites that differ from the on-target site by one or two nucleotides (Supplementary Fig. 17a), whereas off-target edits were induced with low (~1.0%)

frequencies by the *ND2* mitoZFD at a homologous site with a one-nucleotide mismatch (Supplementary Fig. 17b). Use of the QQ variant pair reduced the off-target edit frequency to 0.4% at this site (Supplementary Fig. 17b)."

Supplementary figure 17. Potential off-target sites of mitoZFD in nuclear DNA.

We also have included Fig. 5e, showing an 8.2-fold improvement in the specificity

ratio when the QQ variant is used instead of the WT form.

Fig. 5e, The specificity ratio was calculated by dividing (average editing frequency at on-target Cs) by (average editing frequency at off-target Cs).

editing frequency at off-target Cs).

5. The Split-DddAtox orientation of the ZFD architecture should be labeled, which may be more helpful for understanding “Left-G1333-N + Right-G1333-C” or “Left-G1333-C + Right-G1333-N”.

Response: We followed the notation style (“Left-G1333-N + Right-G1333-C” or “Left-G1333-C + Right-G1333-N”) used in Mok et al. (Nature 583, 631 (2020)), who reported on the development of DdCBEs for the first time. We have added Supplementary Fig. 4e to provide information on ZFD constructs.

e DddA_{tox} split

e-1. CC config. : Left-G1333-N + Right-G1333-C

e-2. CC config. : Left-G1333-C + Right-G1333-N

e-3. CC config. : Left-G1397-N + Right-G1397-C

e-4. CC config. : Left-G1397-C + Right-G1397-N

Supplementary Fig. 4e,

Four ZFD constructs, in which different DddA_{tox} split orientations are used with the left and right zinc fingers.

6. To make ZFD, the authors deleted one or two zinc fingers and added a few zinc fingers. Please provide more detailed information or references.

Response: We have added Supplementary Fig.3 to show how we make ZFDs by modifying ZFPs in pre-characterized ZFNs. We have also added “ZFD design scheme” in the Methods section as follows:

ZFD design scheme

ZFPs used for the construction of ZFDs were prepared in two ways: 1) reengineering of ZFPs in pre-characterized ZFNs, and 2) de novo assembly. In the first approach, new ZFPs were made by removing or adding zinc fingers in pre-characterized ZFPs (Supplementary Fig. 3.) to make ZFDs that operate on a spacer of more than 7 bp in length. In the second approach, ZFPs were assembled from scratch using a publicly available zinc finger resource³⁶. In this resource, 33 zinc fingers were recommended for use. Sequences encoding ZFPs were fused those encoding split DddA_{tox} to produce ZFDs from expression plasmids (Supplementary Fig. 4 and 7).

Supplementary Figure 3.

Supplementary figure

3. Designing the ZFD pairs by modifying previously-characterized ZFNs.

Supplementary Figure 4

Supplementary figure 4.
Possible ZFD architectures and DddA_{tox} orientations.

Supplementary Figure 7

Supplementary figure 7.
Possible mitoZFD architectures.

7. More information on efficiency and off-target effects on the nuclear DNA or mitochondrial DNA is helpful for the discussion.

Response: We have now added the following paragraph in the Discussion section:

“We expect that ZFDs can be further engineered to improve their efficiency and specificity. Here, we showed that use of ZFP variants and ZFD mRNA can reduce off-target activity. DddA_{tox} can also be engineered to avoid ZFD off-target mutations. ZFDs with enhanced efficiency and precision could pave the way for correcting pathogenic mitochondrial DNA mutations in human embryos, fetuses, and patients.”

Reviewers' Comments:

Reviewer #1:

Remarks to the Author:

The authors have addressed the concerns raised by the reviewers to ensure that their ZFD can be used with confidence by users. The quality of the paper has also been improved sufficiently to be published.

Reviewer #2:

Remarks to the Author:

The authors have addressed all of my concerns